# Deep learning-based prediction of major arrhythmic events in dilated cardiomyopathy: A proof of concept study

**Mattia Corianò** [1]ʘ, **Corrado Lanera** [2]ʘ, **Laura De Michieli** [1], **Martina Perazzolo Marra** [1], **Sabino Iliceto** [1], **Dario Gregori** [2], **Francesco Tona** [1] *

**1** Department of Cardiac, Thoracic, Vascular Sciences and Public Health, Padova, Italy, **2** Department of Cardiac Thoracic Vascular Sciences and Public Health, UBEP, Padova, Italy

ʘ These authors contributed equally to this work.
* francesco.tona@unipd.it

## Abstract

Prediction of major arrhythmic events (MAEs) in dilated cardiomyopathy represents an unmet clinical goal. Computational models and artificial intelligence (AI) are new technological tools that could offer a significant improvement in our ability to predict MAEs. In this proof-of-concept study, we propose a deep learning (DL)-based model, which we termed Deep ARrhythmic Prevention in dilated cardiomyopathy (DARP-D), built using multidimensional cardiac magnetic resonance data (cine videos and hypervideos and LGE images and hyperimages) and clinical covariates, aimed at predicting and tracking an individual patient's risk curve of MAEs (including sudden cardiac death, cardiac arrest due to ventricular fibrillation, sustained ventricular tachycardia lasting $\geq$30 s or causing haemodynamic collapse in <30 s, appropriate implantable cardiac defibrillator intervention) over time. The model was trained and validated in 70% of a sample of 154 patients with dilated cardiomyopathy and tested in the remaining 30%. DARP-D achieved a 95% CI in Harrell's C concordance indices of 0.12–0.68 on the test set. We demonstrate that our DL approach is feasible and represents a novelty in the field of arrhythmic risk prediction in dilated cardiomyopathy, able to analyze cardiac motion, tissue characteristics, and baseline covariates to predict an individual patient's risk curve of major arrhythmic events. However, the low number of patients, MAEs and epoch of training make the model a promising prototype but not ready for clinical usage. Further research is needed to improve, stabilize and validate the performance of the DARP-D to convert it from an AI experiment to a daily used tool.

## Introduction

Dilated cardiomyopathy (DCM) is characterized by left ventricular (LV) or biventricular dilation and systolic dysfunction unexplained by coronary artery disease (CAD) or abnormal loading conditions [1, 2]. The aetiology of DCM represents a tangle where a genetic predisposition interacts with extrinsic factors, resulting in a wide spectrum of phenotypes with different

**Funding:** The authors received no specific funding for this work.

**Competing interests:** The authors have declared that no competing interests exist.

natural histories and arrhythmic risks. Therefore, the true prevalence is difficult to evaluate, estimated at 1 in 2700 individuals [3, 4]. The five-year mortality rate ranges between 21% and 28%, with a relevant amount of major arrhythmic events (MAEs), particularly sudden cardiac death (SCD), the incidence of which stands at approximately 12%, accounting for 25–35% of all deaths [5]. Discrimination between patients at a high or low risk for MAE is challenging. Previously, clinicians took into account the value of LV ejection fraction (LVEF) and "New York Heart Association" (NYHA) class for risk stratification [6]. At present, recent findings suggest an important role of cardiac magnetic resonance (CMR), in particular regarding the presence of late gadolinium enhancement (LGE), for the evaluation of arrhythmic risk [1, 7]. However, risk stratification in DCM still lacks accuracy, and a more integrated approach that combines CMR findings with patient characteristics is needed [8]. Computational models and artificial intelligence (AI) are new technological tools that could offer a significant improvement in our ability to predict MAEs. For this purpose, AI algorithms were tested in ischaemic heart disease, reaching good performance in event prediction [9, 10]. Although AI could represent a fundamental change in future decision-making about the aforementioned prediction problem, such an approach has not been widely tested in DCM [11]. Wu et al. [12] first tested a random forest statistical method for risk assessment for ventricular arrhythmias in a population of ischaemic and nonischaemic cardiomyopathies by incorporating clinical covariates and one-dimensional CMR variables. They identified the most predictive variables of MAEs, thus enhancing how AI overperforms regression methods for risk prediction. However, CMR data were manually extracted by two clinicians, and the model did not estimate individual patient times to MAE. Recently, Popescu et al. [9] proposed a deep learning (DL) model that learns from raw clinical imaging data (LGE CMR images only) as well as from clinical covariates, offering a patient-specific probability of MAEs at all times up to 10 years.

To the best of our knowledge, we present here a DL technology extending all the current survival models for the prediction of MAE risk in patients with DCM, which we termed Deep ARrhythmic Prevention in DCM (DARP-D). Our approach embeds dense, convolutional, and convolutionally recurrent neural networks (NNs) [13, 14], learning directly from nonundersampled original raw 2D standard, 3D space-series, 3D time-series, and 4D space-time-series images, together with flat 1D clinical baseline covariates to estimate individual patient risk scores for MAEs.

## Methods

### Study cohort

We retrospectively collected data from consecutive patients referred to the Cardiology Department of the University Hospital of Padua from June 2002 to November 2019 with a diagnosis of DCM. The diagnosis was based on the 1995 World Health Organization/International Society and Federation of Cardiology criteria [15]. Inclusion criteria were as follows: depressed LVEF systolic function (<50%); an angiographic study showing the absence of flow-limiting CAD (defined as ≥50% luminal stenosis on coronary angiography); the absence of either valvular or hypertensive heart disease and congenital heart disease; and patients who had undergone a CMR examination. Exclusion criteria were acute myocarditis in the previous 6 months, other cardiomyopathies (hypertrophic, arrhythmogenic, Takotsubo, restrictive, peripartum), and infiltrative heart disease.

This study was conducted in accordance with the principles of the Declaration of Helsinki and was approved by the Ethics Committee for Clinical Trials of the Province of Padua—Italy (CESC code: 356n/AO/23). Data collection started on 17th of April 2023. Given the retrospective, observational, non-interventional, nature of the study, patients were not asked for a

specific informed consent. All personal identifiers have been removed or disguised to protect the confidentiality and privacy of the participants.

## Baseline features

Baseline data on demographics, clinical characteristics, medical history, medications, lifestyle habits and cardiac test results were collected.

## Follow-up

The follow-up data were obtained by reviewing medical records, routine device interrogation for patients who underwent device implantation, direct interviews during office visits, and telephone contact with the patient or a close family member. The study outcome was a combined endpoint of MAEs, including SCD, cardiac arrest due to ventricular fibrillation, sustained ventricular tachycardia lasting ≥30 s or causing haemodynamic collapse in <30 s, and appropriate implantable cardiac defibrillator (ICD) intervention. SCD was defined, according to the most recent recommendations, as a sudden natural death presumed to be of cardiac cause that occurs within 1h of onset of symptoms in witnessed cases and within 24h of last being seen alive when it is unwitnessed [1]. Event data were censored at 8 years after enrolment or at the time of death, MAE, cardiac transplant or LV assist device implantation or loss to follow-up.

## CMR examination

The CMR images were acquired using a 1.5-T scanner (Magnetom Avanto, Siemens Healthineers, Erlagen, Germany) using dedicated cardiac software, phased-array surface receiver coil and electrocardiogram triggering. The exact software version for the device cannot be precisely ascertained retroactively. For our purpose, we considered steady-state free precession sequence cine and T1-weighted LGE images, which were acquired in multiple short-axis (SAx) and 3 long-axis (LAx) planes. Owing to the retrospective nature of the data collection, for each patient, a different number of images for each plane were obtained, resulting from different repetition times and slice thicknesses. The contrast agent used was 0.20 mmol/kg gadobutrol (Gadovist$^{TM}$), and the scan was captured 8 to 15 min after injection. The most commonly used sequence was inversion recovery fast gradient echo pulse, with an inversion recovery time typically starting at 250 ms and adjusted iteratively to achieve maximum nulling of normal myocardium. The images were evaluated separately by 2 observers (M.C., M.P.M.), and those with extensive artefacts were excluded. LGE-LAx images were collected in standard image format as png files, and multiple LGE-SAx images, cine-LAx, and multiple cine-SAx sequences of images were collected in standard video format as avi files.

## Data preparation

The inputs to our model were the unprocessed CMR images, either LGE-SAx, cine-LAx, and cine-SAx sequences, LGE-LAx images, and the clinical covariates listed in Table 1. The training target was the individual log-risk score component for the Cox proportional hazard function [16].

For a fully detailed description of the preprocessing phase, see **S1 Appendix.**

## CMR images

CMR images were differentiated according to the number of dimensions that characterized them: hypervideo cine-SAx sequences were composed of 3 spatial dimensions (i.e., width,

**Table 1. Clinical covariates.**

| Covariate | Overall (n = 154) | Train (n = 76) | Validation (n = 32) | Train + Validation (n = 108) | Test (n = 46) | p value |
|---|---|---|---|---|---|---|
| Age, y | 48.0(38.0–58.0) | 45.0(37.0–57.5) | 49.0(41.0–60.2) | 48.0(38.0–59.0) | 49.0(36.0–57.0) | 0.591 |
| Male, n (%) | 108(70) | 52(68) | 22(69) | 34(31) | 34(74) | 0.789 |
| Height, m | 1.7(1.7–1.8) | 1.7(1.6–1.8) | 1.7(1.7–1.8) | 1.7(1.7–1.8) | 1.7(1.7–1.8) | 0.480 |
| Weight, kg | 79(67.0–88.0) | 83.0(61.0–88.5) | 78.0(70.7–86.2) | 79.0(65.0–87.0) | 80.0(72.0–94.0) | 0.504 |
| Dyslipidaemia, n (%) | 35(26) | 19(29) | 4(14) | 23(24) | 12(29) | 0.243 |
| Arterial hypertension, n (%) | 52(38) | 22(34) | 13(45) | 35(37) | 17(41) | 0.539 |
| Smoker | | | | | | 0.066 |
| Current smoker, n (%) | 31(23) | 17(26) | 7(24) | 24(25) | 7(17) | |
| Ex-smoker | 17(13) | 13(20) | 1(3) | 14(15) | 3(7) | |
| Diabetes mellitus, n (%) | 18(13) | 4(6) | 4(14) | 8(8) | 10(24) | 0.027 |
| Familial history | | | | | | |
| CAD, n (%) | 18(13) | 7(10) | 4(14) | 11(11) | 7(17) | 0.578 |
| Cardiomyopathy, n (%) | 22(16) | 16(24) | - | 16(17) | 6(15) | 0.013 |
| SCD, n (%) | 7(5) | 5(7) | - | 5(5) | 2(5) | 0.318 |
| COPD, n (%) | 3(2) | 2(3) | - | 2(2) | 1(2) | 0.646 |
| Creatinine, mmol/l | 81.0(69.0–91.2) | 82.0(66.5–91.5) | 84.0(67.7–101.2) | 82.0(69.0–93.0) | 79.0(71.0–89.0) | 0.406 |
| NYHA | | | | | | 0.826 |
| I, n (%) | 85(56) | 41(55) | 16(50) | 57(53) | 28(61) | |
| II, n (%) | 14(9) | 6(8) | 4(12) | 10(9) | 4(9) | |
| III, n (%) | 50(33) | 27(36) | 11(34) | 38(35) | 12(26) | |
| IV, n (%) | 4(3) | 1(1) | 1(3) | 2(2) | 2(4) | |
| NT-proBNP, pg/l | 923(593–2309) | 845(610–2442) | 880(309–1786) | 845(466–1949) | 1009(646–2523) | 0.947 |
| Sinus rhythm, n (%) | 113(86) | 58(88) | 21(78) | 79(85) | 34(87) | 0.428 |
| Atrial fibrillation, n (%) | 20(15) | 8(12) | 7(13) | 15(16) | 5(13) | 0.215 |
| LBBB, n (%) | 56(36) | 36(47) | 9(28) | 45(42) | 11(24) | 0.018 |

CAD: coronary artery disease; COPD: chronic obstructive pulmonary disease; LBBB: left bundle branch block; NYHA: New York Heart Association; SCD: sudden cardiac death. Data are reported as median (1st– 3rd quartile) for continuous variables and as total number (%) for categorical variables. Differences between "train", "validation" and "test" groups were assessed using the Mann–Whitney test for continuous variables and the Pearson chi square test of Fisher exact test for categorical variables. P values < 0.05 were considered statistically significant.

height, and slice) and 1 time dimension; standard video cine-LAx sequences were composed of 2 spatial dimensions (i.e., width and height) and 1 time dimension; standard LGE-LAx images were composed of 2 spatial dimensions (i.e., width and height); hyperimages LGE-SAx sequences were composed of 3 spatial dimensions (i.e., width, height, and slice). Because of the heterogeneity in number of time frames (temporal dimension) and number of slices (spatial dimensions), "null" frames and slices were added to obtain homogeneous hypervideos sequences of 4 dimensions.

## Clinical covariates

Clinical covariates included in the DARP-D are listed in Table 1, and all of them are well known to be independent risk factors for MAEs in DCM [1, 6]. They concern information about demographic features, cardiovascular risk factors, comorbidities, blood tests, functional status and electrocardiographic characteristics.

## Neural network architecture

DARP-D is a supervised multi-input deep neural regression network. It is composed of three main branches trained synergically. Two of them use CMR sequences as input data, while the third one processes clinical data. All three are injected in the main path of the network. CMR branches are mainly powered by pooling, convolutions, and convolutional recurrent architectures, while the clinical and main branches are basically sequences of fully connected dense layers.

The last linearly activated single-node output layer of the network takes the role of the individual nonlinear log-risk score, which is used to evaluate the patient-individual risk curve of MAE.

All the code was developed in R 4.2.2 using the TensorFlow and keras R packages as interfaces to the corresponding TensorFlow and Keras Python deep-learning platforms [17–19]. The targets R-package is utilized to orchestrate and automate the pipeline dependencies and computations [20].

## Images and covariates analysis

Two types of NNs were used together to build DARP-D. Long short-term memory (LSTM) network, a particular type of recurrent neural network (RNN) is able to maintain complex relevant information such as temporal correlations [21–23]. Convolutional neural network (CNN) allows to model complex spatial correlations from the input data [24–26]. In our model, to process 4D and 3D cine-CMRs, we adopted ConvLSTM architectures [14]. The final architecture proposed concatenated all four cine-CMRs in a multidimensional array of fixed dimensions and then processed with a ConvLSTM. At the same time, all four LGE-CMRs were concatenated in another multidimensional array of fixed dimensions, and then processed with a CNN. Afterwards, all multidimensional arrays received a progressive reduction of dimensions until they were merged and flattened to a linear (1 dimension) array. A similar process of flattening was applied to the clinical covariates, and the two arrays were concatenated together. The resulted array was processed in order to obtain on output (x), which was used as a coefficient of the Cox hazard function ($\hat{h}_{DARP-D}(x)$) [27, 28].

## Survival model

We propose an innovative per-patient survival model that expands the family of so-called nonlinear Cox models powered by modern DL techniques [9, 29, 30]. The DL architecture permits processing in a unique heterogeneous network the uncompressed not-interpolated raw time-dependent 4D (cine-SAx) hypervideos, 3D (cine-LAx) videos, 3D (LGE-SAx) hyperimages, and 2D (LGE-LAx) images, together with baseline patient covariates. The process allows direct end-to-end estimation from CMRs and clinical data to the individual nonlinear log-risk function $h(x)$ as $\hat{h}_{DARP-D}(x)$ [16].

## Performance metrics

The performances of the models were evaluated using two measures. To evaluate the model's risk discrimination ability, Harrell's C-index is used, considering predicted network outputs as patient-specific log-risk scores [31]. The second was the area under the curve (AUC) for the model to be considered as a classification for a within 5-year MAE binary outcome.

## Training and testing

Out of 154 patients, the model was trained on a random sample of 76 patients and optimized using a validation subset of 32 (~30% of the 108 used training data). Performance was tested in the out-of-training test set, counting the other 46 patients (approximately 30% of the total).

Considering the proof-of-concept nature of the study, DARP-D was implemented with 5 epochs of training in the first training vs. validation set stage to set the base hyperparameters, i.e., batch size, regularization, to allow the computation to fit in memory, converges, and trends to improve on validation set, in order to evaluate the feasibility of that kind of model without exceeding in computational time. Next, we continued to train the model from both training and validation for the other 25 epochs, validated on the hold-out test set. For further technical information, see **S1 Text.**

## Statistical analysis

Baseline characteristics are summarized as the median (1st– 3rd quartile) for continuous variables and n (%) for categorical variables.

Baseline covariates were reported as median values for continuous variables and as frequencies for categorical variables. Time to first MAE event, loss to follow-up or death was calculated from the baseline CMR to compute the follow-up time for survival analyses.

## Results

### Cohort characteristics and follow-up

The overall cohort consisted of 154 patients, with a median age of 49 years and a median follow-up time of 60 months. The baseline characteristics of the cohort are shown in Table 1. In summary, males were more represented (71%), and the most common risk factor was arterial hypertension (37%), followed by smoking habits (35%). A positive familial history of cardiomyopathy and SCD was present in 17% and 5% of patients, respectively. The majority of patients presented few symptoms (NYHA I, 88%) and were in sinus rhythm (86%). All patients took heart failure (HF) medication, mainly β-blockers and angiotensin-converting enzyme inhibitors. CMR measurements showed a median left ventricular end diastolic volume index (LVEDVi) of 137 ml/m2 and a LVEF of approximately 28%, while the median right ventricle (RV) end-diastolic volume index (RVEDVi) and ejection fraction (RVEF) were 56 ml/m2 and 52%, respectively. Data about medication use, CMR measurements and follow-up are listed in S1 Table. Overall, after a median of 6 years of follow-up, MAE occurred in 20 patients, with an incidence rate of 12% at 6 years after enrolment. Concerning the non-MAE endpoint, there were 12 all-cause deaths, 12 patients sustained a heart transplant, and one received an LV assist device (incidence rate of 22% at 6 years). No differences were observed between the validation and test subgroups, except for a family history of cardiomyopathy, which was more frequent in the validation subgroup, and of left bundle branch block, which was more frequent in the test subgroup. Fig 1 reports event-free survival at 8 years of the overall population and of the training, validation, and test subgroups. By the end of follow-up, 15% of all patients had experienced a MAE. The log rank test of the three curves showed that they were not significantly different (p = 0.088).

### DARP-D overview

The arrhythmia risk assessment algorithm in DARP-D consists of a supervised multi-input deep neural regression network ingesting multidimensional CMRs and baseline clinical information trained synergically to predict patient-specific probabilities of MAE at future time points. As shown in Fig 2, the model consists of three main branches of a common network, which implements the MAE log-hazard individual function and returns the current individual MAE log-hazard score based on current CMRs and clinical situation as output. Subsequently,

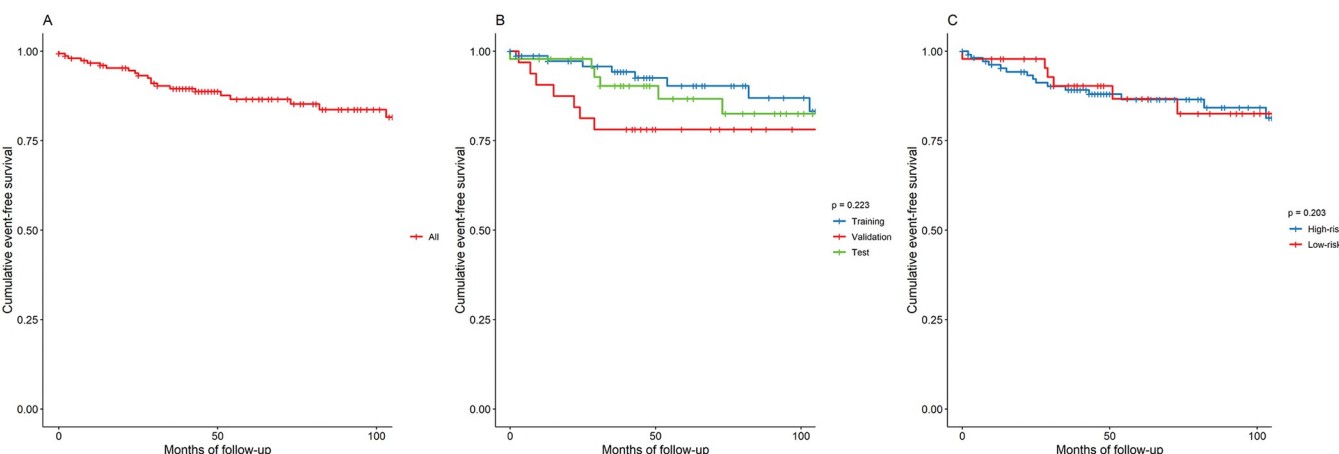

**Fig 1. Event-free survival from major arrhythmic events.** Event free survival at 96 months from major arrhythmic event, defined as sudden cardiac death, cardiac arrest due to ventricular fibrillation, sustained ventricular tachycardia lasting $\geq$30s or causing hemodynamic collapse in <30s, appropriate implantable cardiac defibrillator intervention. Tick marks indicate censored data. **A**. Overall event free survival. **B**. Event free survival for train, validation and test subgroups and log-rank test. **C**. Event free survival and log-rank test for patient at high and low risk of event in the test set. Risk of event is directly estimated by the model from the individual nonlinear log-risk function $\hat{h}_{DARP-D}(x)$, where x is the output of the single-neuron last layer output of the neural network.

Cox survival analysis uses the patient network outputs to estimate the time-dependent population base hazard function to obtain a patient-specific probability of MAE at any time point.

## DARP-D risk prediction performance

The DAPR-D was developed, internally validated, and tested using data from our cohort of 154 patients with DCM. Its performance was evaluated using Harrell's concordance index (c-index) [31]—range is [0, 1], higher scores are better—and areas under the receiver operator characteristic curve (AUROC) evaluated at years 1, 2, 3, 5 and 8. Currently, the DARP-D still has a quite low and unstable concordance index on the hold-out test set (0.12–0.68) (Table 2). On the other hand, learning curves report both training and validation performances in a high improving phase, showing that overfitting is still under control and far from being an issue, meaning that further training epochs and data would critically improve the model (Fig 3).

The model risk discrimination abilities at all times, represented by the AUROC evaluated at years 1, 2, 3, 5, and 8, were 84%, 84%, 64%, 64% and 53%, respectively, on the test set (S1 Fig).

## Discussion

### General considerations

We present an innovative approach to predict MAEs, termed DARP-D, which uses a deep NN "survival" model for the risk assessment of fatal arrhythmia in DCM. The model was trained using two types of input data, CMR sequences and clinical covariates. The choice of the clinical variables was made considering the current knowledge about risk factors and comorbidities associated with MAEs. In fact, all variables are well recognized independent factors of MAEs in DCM [32]. Moreover, our cohort showed baseline characteristics that were similar to other cohorts represented in clinical trials and prospective registers of DCM [33–35]. This similarity was marked by the outcome analysis, with an analogue percentage of MAEs and overall mortality occurring during the follow-up. Concerning CMR sequences, the rationale for including cine videos and hypervideos comes from the well-established knowledge that LVEF, considered a surrogate of cardiac contractility, strongly correlates with arrhythmic prognosis; thus,

## CMR IMAGES BRANCHES

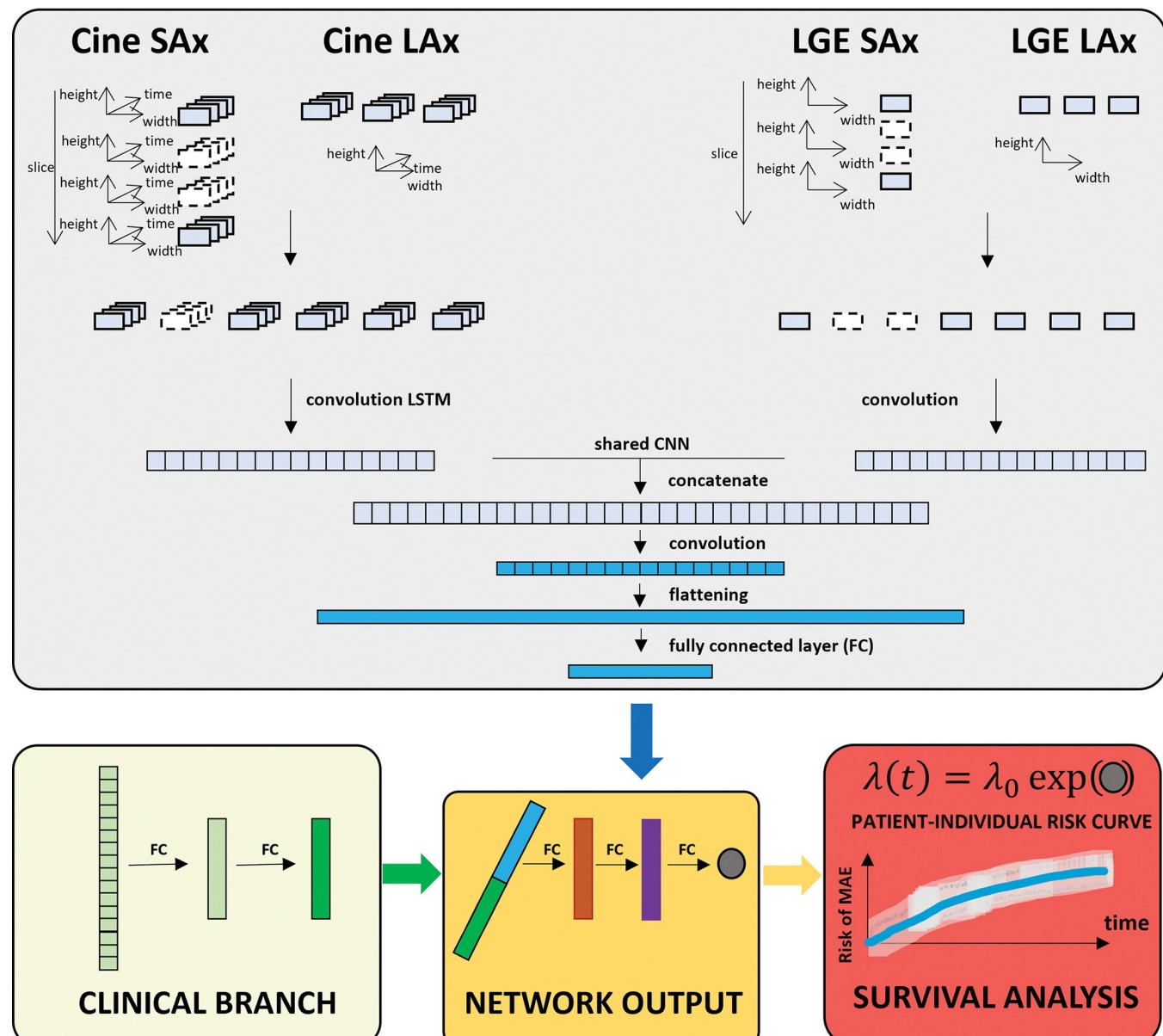

**Fig 2. Schematic overview of DARP-D.** Top panel (grey) shows the first two branches of the model, which use, respectively, unprocessed cine- and LGE-CMR data. Cine-CMR hypervideos are taken as input by a 4D and 3D convolutional long term short memory network constructed using an encoder architecture. LGE-CMR hyperimages are taken as input by a 3D and 2D convolutional neural network. These two branches convey in a common network (shared convolutional neural network) determining a one dimensional output of shape 8. Left bottom panel (yellow) shows the covariate branch, which consists of two consecutive sets of 8 fully connected networks producing a tensor of shape 8. Bottom central panel (orange) shows the final single fully connected network that uses as inputs the tensors from CMR and covariate networks to give a one dimensional output of shape 1. Right bottom (red) panel shows the survival model, where the output of shape 1 is used to directly estimate the individual nonlinear log-risk function $\hat{h}_{DARP-D}(x)$.

an analysis of the entire cardiac cycle allows us to take into account systolic function [1, 36]. Furthermore, LGE images were included because of the growing evidence that the extent, location and pattern of LGE correlate with MAE in a nonlinear relationship [8, 35].

The relevance, in terms of outcome prediction, of merging CMR images and clinical covariates in a DL model was enhanced in a recent study by Popescu et al. [9]. They showed that the

**Table 2. Performance of DARP-D.**

| Set | Harrell's C | SD | Lower 95% CI | Upper 95% CI |
|---|---|---|---|---|
| Training | 0.558 | 0.228 | 0.330 | 0.786 |
| Validation | 0.325 | 0.195 | 0.130 | 0.520 |
| Test | 0.399 | 0.278 | 0.121 | 0.677 |

accuracy of a survival DL-based model increased by adding clinical covariates to CMR acquisitions, resulting in a better prediction of MAEs in ischaemic cardiomyopathy. Starting from this assumption, we built the DARP-D with the aim of improving the risk stratification of MAEs in DCM, a problem that currently represents a clinical unmet goal. In this study, our model fit together CMR sequences and clinical covariates, where we used both cine and LGE sequences for training. Our approach represents the first examples of a DL architecture where motion (cine videos and hypervideos), tissue characterization (LGE images and hyperimages), and clinical variables concur to the risk stratification of MAEs in DCM. The analysis of cine hypervideos represents a novelty in the prognostic field of cardiomyopathies. Indeed, the state-of-the-art DL analysis of cine sequences consists of a multiview motion estimation network for 3D myocardial motion tracking [37]. In contrast, our work started with a different aim, that is, to consider cardiac motion as a patient characteristic that concurs with other characteristics (LGE and clinical variables) in the arrhythmic prognosis.

DARP-D achieved unstable performance possibly because of the use of a relatively small dataset and the low training epochs reached. A concern with DL on smaller datasets is overfitting, which manifests itself as high performance during training (good fit) but poor performance when applied to a new test set. To speed up the training, control the overfitting, and protect from exploding and vanishing weights, after each layer of the network is described, stacked batch normalization, activity regularization, and drop-out are performed. The efficacy of this approach is reflected in the uniform improvement trends on the validation set, as shown by the learning curves in Fig 3. Nonetheless, the supposed improvement in performance is theoretical and needs to be proven with further research before translation into clinical practice.

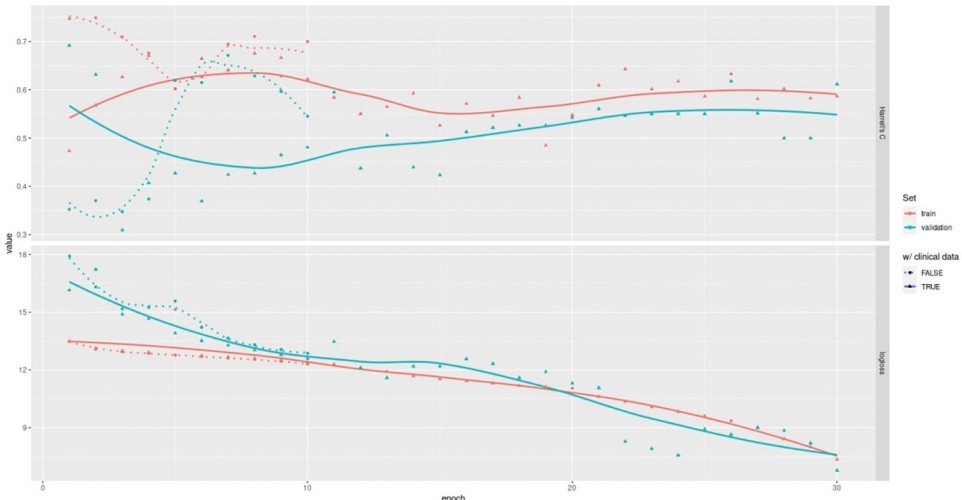

**Fig 3. DARP-D learning curve.** Epoch-series (x-axis) vs. performance (y-axes) learning curves for the DARP-D first stage of training (30 epochs): training (red) vs. validation (green) sets. C (below) reports Harrell's C concordance index, loss (above) reports the progression for the log partial likelihood for the Cox survival model.

The performance of DARP-D needs to be contextualized in the proof-of-concept nature of our study. On the one hand, considering the model from a clinical point of view, DARP-D is not ready for clinical practice because of its low performance, as shown by Harrell's C and AUROCs. On the other hand, considering DARP-D from a technical point of view, its potentiality is unquestionable. In fact, we built a model that was able to analyze different kinds of data (i.e., regarding nature, dimensionality and frequency of the data), and the process below (i.e., data acquisition, dimension flattening, convolutional recurrent NNs and per-patient survival model) works straightforwardly, such as the training-validation-test processes. Building a model able to be directly translated into clinical practice was beyond the scope of this research, which is the reason why the training process was conducted up to 30 epochs only, and more advanced and tailored network components were not considered. A follow-on working prototype, ready to be translated into clinical practice, will be the object of future research and the subject of stronger validation stages.

### Survival analysis and patient-specific survival curves

We propose a per-patient survival model based on modern deep-learning techniques capable of processing conjointly uncompressed noninterpolated raw time-dependent 4D hypervideos, 3D videos, 3D hyperimages, 2D images and baseline patient covariates to estimate the individual nonlinear log-hazard function.

DARP-D opens perspectives in patient-specific differential MAE risk analyses directly comparing both CMRs and clinical factors from an integrated model, expanding to video and image tools such as odds ratios, reserved to clinical data only up to now. With DARP-D, it would be possible to set direct comparisons for patient evolution of MAE risk across successive follow-up, empowering the synergistic evolution of both heart dynamics, as captured by the CMRs, and the corresponding changes in the other clinical measures.

### Limitations

Our study has several limitations. The first concern is regarding the study cohort, which consisted of only 154 patients utilized for training, validation, and testing. When developing a DL model, it is advisable to ensure that the sample size suffices to enable reliable prediction in new individuals. While a pre-specified sample size cannot be calculated a priori, it should be large enough to develop a model that proves reliable when applied to new individuals. From a general perspective, the minimally required sample size for a prediction model is higher than that needed for a regression-based model and it depends on the prevalence in the target population, the predictive value of the features, and the complexities of the features [38, 39]. Practically speaking, several hundreds of patients are usually required. This remarks that DARP-D, at present, is a prototype and needs testing in a larger dataset capable of representing the wide heterogeneity of the DCM population. Moreover, an external validation is needed to confirm the potential impact of the DARP-D in predicting MAEs in different cohorts of patients.

Second, our project aimed to develop a future model capable of supporting clinicians to improve therapeutic strategies for fatal arrhythmic event prevention, such as device implantation. It is important to consider that, for DCM, current guidelines recommend ICD implantation for primary prevention of SCD after 3 months of optimal medical therapy (OMT). OMT is considered as the using of all the "four pillars" suggested by HF guidelines (i.e. beta-blockers, angiotensin converting enzyme inhibitor or angiotensin receptors blockers or angiotensin receptor/neprilysin inhibitor, mineralocorticoid receptor antagonist and gliflozin) and, when appropriate, the implant of a cardiac resynchronization therapy device [6]. Our cohort encompasses patients evaluated in a substantial period of time (from 2002 to 2019); during this

period, new drug therapies were introduced in the HF treatment strategy, such as sacubitril and gliflozin, but a very low percentage of patients in our cohort took any of these medications. This suggests the need for external validation to enhance the performance of the DARP-D in a more recent cohort.

Third, the preprocessing step focuses on a dimensional reduction of hypervideos and hyperimages but not on cardiac segmentation. CMR images taken as input by the CMR-NNs were not automatically segmented to include myocardium-only raw intensity values. Theoretically, this does not represent a true limit itself, even if many previous studies applied such a preprocessing step. It would be interesting to determine if image segmentation increases the accuracy of the model. Further research will follow to investigate this possibility.

Fourth, the number of epochs of training was low compared to other research in the fields of DL application in cardiology. In this proof-of-concept study, we did not aim to build a model ready for large-scale use or with high performance. Instead, our study showed that a more detailed risk stratification, based on a DL analysis of cine hypervideos, LGE images and clinical covariates, is feasible and offers critically promising results in terms of risk score concordance and accuracy of event prediction. If confirmed by further research, a similar approach could be used for other forms of cardiomyopathies, such as hypertrophic and arrhythmogenic cardiomyopathy. Therefore, DARP-D was implemented with a maximum of 30 epochs, and more robust training will follow in the future.

Fifth, the DARP-D was trained only to predict MAEs without considering competing risk. Other possible causes of death may be related to a non-MAE event (e.g., death due to heart failure) or to other MAE not directly associated with the condition under investigation. In our study, the cohort was selected based on the presence of specific structural abnormalities (LV dilation and reduction of EF) and the absence of other structural abnormalities (other forms of cardiomyopathies). It is well known that there are other conditions associated with MAE that do not usually exhibit detectable structural abnormalities with CMR. Brugada syndrome (BS) and catecholaminergic polymorphic ventricular tachycardia (CPVT) can be considered as two significant examples. Both syndromes can cause SCD, and their diagnosis can be challenge as they typically present no alteration on CMR [40, 41]. In our study, we retrospectively selected our cohort by reviewing anamnestic reports, and all patients with DCM that we analyzed did not have any mention of a concurrent diagnosis of BS or CPVT, nor did they have a previous period of monitoring with implantable device such as loop recorder. Nevertheless, no other diagnostic tests were reported to have been performed to exclude these form of channelopathies, and this bias could have influenced the result.

Considering the aim of our study, this does not represent an obstacle to our purpose. However, this aspect needs to be taken into consideration in further studies, where the clinical usefulness of the DARP-D will be the core of the research. In fact, this is a crucial clinical point because the benefit of preventing an arrhythmic event (maybe implanting an ICD) should be balanced with the life expectancy of patients with DCM, who are at high risk of other non-MAE causes of death.

Another consideration pertains to the evaluated endpoint. We considered a composite endpoint of SCD and SCD equivalents, including appropriate ICD intervention. All patients with an ICD enrolled in this study had a transvenous device, and therefore, the presence of appropriate antitachycardia pacing (ATP) therapy was included in the MAE endpoint. Currently, the increasing use of subcutaneous ICDs (S-ICD) raises questions about the efficacy of such devices in cardiomyopathies and how to evaluate clinical arrhythmic endpoints. Although no clinical trial was conducted specifically in the setting of cardiomyopathies, substantial evidence suggests that S-ICD efficacy appears non-inferior to transvenous ICDs in terms of preventing SCD and all-cause mortality [42, 43]. Moreover, the inability of S-ICD to perform ATP was

not associated with a higher risk of MAE. This implies that assuming a composite endpoint including ATP-appropriate intervention could correspond to a lower incidence of endpoints in future cohorts with patients with S-ICD.

The last limitation regards the interpretability of the DARP-D. This field of AI algorithms is paramount to their broad adoption, and concerns surrounding it are particularly prevalent in healthcare. We did not perform any analysis that could provide more understandable results. Such an analysis will be scheduled to render transparency to the algorithm "black box".

Altogether, the aforementioned limitations do not reduce the value of DARP-D. Rather, they pave the way for further research to improve its prediction ability, to confirm its strength in external cohorts and to make the results more understandable.

## Conclusion

In this study, we presented a DL technology, DARP-D, trained on a cohort of patients with DCM and capable of learning from clinical covariables and CMR hypervideos and hyperimages, returning a specific per-patient time-dependent risk of MAEs. Our approach could represent a fundamental change in the prevention of arrhythmic death in DCM. However, the low number of patients, MAEs and epoch of training make the model a promising prototype but not ready for clinical usage. Further research is needed to improve, stabilize, and validate the performance of the DARP-D to convert it from an AI experiment to a daily used tool.

## Supporting information

**S1 Appendix. Extended methods.**
(PDF)

**S1 Text. Technical aspects.**
(PDF)

**S1 Fig. Survival study of major arrhythmic event at different time point.** Receiver operator characteristic curves (ROC) for years 1, 2, 3, 5 and 8 for the internal validation and test sets, with the respective areas under the curve (AUROC). Predicted outcomes are based on the estimated survival probability at the respective time points as computed from the survival probability function.
(PDF)

**S1 Table. Baseline and CMR characteristics and follow-up.**
(PDF)

## Author Contributions

**Data curation:** Mattia Corianò, Corrado Lanera, Laura De Michieli.

**Formal analysis:** Corrado Lanera.

**Funding acquisition:** Sabino Iliceto.

**Investigation:** Mattia Corianò, Corrado Lanera.

**Methodology:** Mattia Corianò, Corrado Lanera, Francesco Tona.

**Project administration:** Francesco Tona.

**Resources:** Sabino Iliceto.

**Supervision:** Martina Perazzolo Marra, Dario Gregori, Francesco Tona.

**Visualization:** Martina Perazzolo Marra, Dario Gregori, Francesco Tona.

**Writing – original draft:** Mattia Corianò, Corrado Lanera.

**Writing – review & editing:** Francesco Tona.

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
