## [Decision Letter · Decision Letter 0]

30 Nov 2023

PONE-D-23-20202Deep learning-based prediction of major arrhythmic events in dilated cardiomyopathyPLOS ONE

Dear Dr. Corianò,

Thank you for submitting your manuscript to PLOS ONE. After careful consideration, we feel that it has merit but does not fully meet PLOS ONE’s publication criteria as it currently stands. Therefore, we invite you to submit a revised version of the manuscript that addresses the points raised during the review process.

ACADEMIC EDITOR: the authors propose an interesting study. Some relevant limitations weaken the conclusions. All issues raised by reviewers are required.=============================

We look forward to receiving your revised manuscript.

Kind regards,

Vincenzo Lionetti, M.D., PhD

Academic Editor

PLOS ONE

Journal Requirements:

2.Thank you for stating the following financial disclosure:

"NO"

"NO"

6. Please upload a new copy of Figures 1, 2 and 3 as the detail is not clear. Please follow the link for more information: " ext-link-type="uri" xlink:type="simple">https://blogs.plos.org/plos/2019/06/looking-good-tips-for-creating-your-plos-figures-graphics/"
https://blogs.plos.org/plos/2019/06/looking-good-tips-for-creating-your-plos-figures-graphics/

Additional Editor Comments:

The authors shoul carefully highlight major limitations of the study (i.e.: sample size). Moreover, title should be modified as follows: "Deep learning-based prediction of major arrhythmic events in dilated cardiomyopathy: a proof of concept study".

Reviewers' comments:

Reviewer's Responses to Questions

Comments to the Author

1. Is the manuscript technically sound, and do the data support the conclusions?

Reviewer #1: Yes

Reviewer #2: Yes

Reviewer #3: Partly

2. Has the statistical analysis been performed appropriately and rigorously? 

Reviewer #1: Yes

Reviewer #2: Yes

Reviewer #3: Yes

3. Have the authors made all data underlying the findings in their manuscript fully available?

Reviewer #1: No

Reviewer #2: Yes

Reviewer #3: Yes

4. Is the manuscript presented in an intelligible fashion and written in standard English?

Reviewer #1: Yes

Reviewer #2: Yes

Reviewer #3: Yes

5. Review Comments to the Author

Reviewer #1: In the manuscript, "Deep learning-based prediction of major arrhythmic events in dilated cardiomyopathy," Dr. Corianò and colleagues attempt to create a predictive model of major adverse events in patients with dilated cardiomyopathy. Using MRI data, both cine and LGE, combined with clinical data, analyzed through artificial intelligence they seek to answer the unresolved question of predicting the risk of cardiovascular death in this population.

The paper is well written and great space has been given to describing the deep-learning model. This more engineering-oriented section is difficult for a medical population to read, and perhaps summarizing it in simpler terms and leaving the more technical description in the supplemental material would have made the work more reader-friendly

The study, while conceptually well structured, did not produce clinically usable results. Nevertheless, the major effort of the work was precisely to create a predictive model that could be implemented over time by increasing the population analyzed.

For this very reason, I believe it would have been useful to create the model on the experimental center population and validate it on a control population provided by other centers.

Increasing the case population and performing multicenter work would have strengthened the hard work done in an important way.

However, the authors are to be congratulated for their attempt at increasingly AI-driven precision medicine, which will surely be the direction for the foreseeable future.

Reviewer #2: The manuscript described a deep learning-based prediction of major arrhythmic events in dilated cardiomyopathy. The aim of the study and the results are well described. The bibliography is accurate and recent articles are citated.

Reviewer #3: This topic is interesting, representing the future: however, predicing an individual patient’s risk curve of major arrhythmic events is extremely difficult for the authors due to the very low number of patients making the model a promising prototype but not ready for clinical usage. I appreciate the big effort, authors should more discuss some points

-“Exclusion criteria were acute myocarditis in the previous 6 months, other cardiomyopathies such hypertrophic, arrhythmogenic, Takotsubo, restrictive, peripartum, and infiltrative heart disease”. Did authors consider concomitant causes of sudden cardiac death with no impact on CMR? How did authors exclude concomitant hidden/intermittent diagnosis especially considering Brugada syndrome or cathecolaminergic VT ( DOI: 10.1016/j.jacc.2018.06.037 and DOI: 10.1111/jce.16045). Please expand this important topic including author’s view and cite 2 suggested references

-The concept of an “optimal medical therapy” should be better clarified. Please discuss

-ICD indications should be more discussed, including potential differences in arrhythmia monitoring between subcutaneous and transvenous devices, due to differences between dilated cardiomyopathy and channelopathies (DOI: 10.1016/j.ijcard.2021.11.029 and DOI: 10.1007/s00380-022-02204-x) Please discuss and cite 2 suggested references

-Was there a role for loop recorder in patient population? Please discuss

6. PLOS authors have the option to publish the peer review history of their article (what does this mean?). If published, this will include your full peer review and any attached files.

Do you want your identity to be public for this peer review? For information about this choice, including consent withdrawal, please see our Privacy Policy.

Reviewer #1: No

Reviewer #2: Yes: Martina Nesti

Reviewer #3: No

---

## [Author Response · Author response to Decision Letter 0]

28 Dec 2023

Mattia Corianò, MD

University of Padova, Department of Cardiac, Thoracic, Vascular Sciences and Public Health, Padova, Italy

Via Niccolò Giustiniani 2

Italy, Padua, 35131

mattia.coriano@studenti.unipd.it

26 December 2023

To:

Vincenzo Lionetti

Plos One Academic Editor

Subject: Response to Reviewers for Manuscript “Deep learning-based prediction of major arrhythmic events in dilated cardiomyopathy”.

Dear Editor,

I hope this letter finds you well. I am writing to express my sincere gratitude for the thoughtful and constructive feedback provided by you and the reviewers on my manuscript. The insightful comments have significantly contributed to the improvement of the manuscript.

I appreciate the time and effort the reviewers invested in reviewing my work. Their feedback has been invaluable, and I have carefully considered each comment in the revision process. In this response, I have addressed each comment and made the necessary revisions to enhance the clarity, validity, and overall quality of the manuscript.

Below, I outline the major comments raised and detail my responses:

Editor: advised highlighting major limitations of the study and emphasizing the proof-of-concept nature of the research in the title.

We appreciate the Editor's insightful feedback on emphasizing the major limitations of our study and underlining the proof-of-concept nature of the research.

In accordance with this constructive advice, we have revised the title of our manuscript. Additionally, we have expanded the discussion in the manuscript to provide a more thorough explanation of the limitations, particularly focusing on the reduced sample size. Changes are at rows 334-343 and are highlighted in red. 

These changes aim to provide a clearer context for readers regarding the exploratory nature of our research and the associated constraints. We believe that addressing these aspects will enhance the transparency and interpretation of our findings.

We sincerely thank the Editor for guiding us in improving the clarity and transparency of our manuscript.

Reviewer #1 provided valuable feedback, suggesting a less technical description of the methods. Moreover, he recommended us to perform a validation of the model by increasing the cohort of patients and performing a multicentre work.

In response to this constructive feedback, we have revised the methods section of the manuscript to offer a more accessible and concise overview. In particular, we have made adjustments to the following subparts: data preparation [106]; CMR images [109-115]; Images and covariates analysis [132 – 142]; Survival model [144 – 149]. The original, more detailed, technical description was added in the Supplementary Material. All changes are highlighted in yellow. 

We thank reviewer #1 for the recommendation to increase the performance of our model. We have planned a retraining-test process of the model with data from a multicentre CMR registry.

Reviewer #2 did not suggest any changes and provided positive feedback on the manuscript.

We sincerely appreciate the time and effort invested by Reviewer #2 in reviewing our manuscript. We are grateful for the positive feedback and are pleased to hear that the content met the expectations. Your encouragement motivates us to continue our efforts in advancing this research.

Reviewer #3 raised questions about whether concomitant causes of sudden cardiac death were considered, particularly those with no impact on CMR. Additionally, there was a query on how concomitant hidden diagnoses, such as Brugada syndrome or catecholaminergic ventricular tachycardia, were excluded, suggesting an expansion of this topic with references. Further clarifications were sought on the concept of "optimal medical therapy," differences in arrhythmias monitoring between subcutaneous and transvenous devices, and the role of a loop recorder in the patient population.

We sincerely appreciate the insightful inquiries raised by Reviewer #3. In response to these constructive comments, we have carefully revised the "Limitation" section of the manuscript. Specifically, we now explicitly state that we investigated the absence of channelopathies by reviewing anamnestic reports. However, it should be noted that no other diagnostic tests were reported to have been performed to exclude these forms of channelopathies, and this potential bias could have influenced the results. 

Furthermore, we have provided a more detailed explanation of the concept of "optimal medical therapy" in accordance with the recommendations outlined in the ESC HF guidelines (2021). 

Regarding the subcutaneous implantable cardioverter-defibrillator (S-ICD), we highly value this feedback. It is a crucial aspect that should be considered in future studies, especially concerning the differences in endpoint incidence based on the inclusion of antitachycardia pacing therapy. This information can significantly impact the discrimination ability of a deep learning model.

As for the role of a loop recorder in our study, we did not identify anamnestic data regarding this information during the review of anamnestic reports. We acknowledge the importance of this aspect and will consider it in our future investigations.

We express our gratitude for the valuable feedback provided, which has significantly contributed to the refinement of our manuscript.

The relevant changes can be found in rows 346 – 351, 369 – 381, 387 – 396, and are highlighted in green. All suggested references were cited. 

I believe that the revisions have effectively addressed the concerns raised by the reviewers, resulting in a manuscript that is stronger and more robust. I am confident that these changes will contribute positively to the overall quality and impact of the paper.

I would like to express my appreciation for the opportunity to revise and resubmit my manuscript to Plos One. I look forward to hearing the final decision and appreciate your consideration of my work.

Thank you once again for your time and for overseeing the review process.

Sincerely,

Mattia Corianò, MD

---

## [Decision Letter · Decision Letter 1]

15 Jan 2024

Deep learning-based prediction of major arrhythmic events in dilated cardiomyopathy: a proof-of-concept study

PONE-D-23-20202R1

Dear Dr. Corianò,

We’re pleased to inform you that your manuscript has been judged scientifically suitable for publication and will be formally accepted for publication once it meets all outstanding technical requirements.

Kind regards,

Vincenzo Lionetti, M.D., PhD

Academic Editor

PLOS ONE

Additional Editor Comments (optional):

Reviewers' comments:

Reviewer's Responses to Questions

**Comments to the Author**

1. If the authors have adequately addressed your comments raised in a previous round of review and you feel that this manuscript is now acceptable for publication, you may indicate that here to bypass the “Comments to the Author” section, enter your conflict of interest statement in the “Confidential to Editor” section, and submit your "Accept" recommendation.

Reviewer #2: All comments have been addressed

Reviewer #3: All comments have been addressed

2. Is the manuscript technically sound, and do the data support the conclusions?

Reviewer #2: Yes

Reviewer #3: Yes

3. Has the statistical analysis been performed appropriately and rigorously? 

Reviewer #2: Yes

Reviewer #3: Yes

4. Have the authors made all data underlying the findings in their manuscript fully available?

Reviewer #2: Yes

Reviewer #3: Yes

5. Is the manuscript presented in an intelligible fashion and written in standard English?

Reviewer #2: Yes

Reviewer #3: Yes

6. Review Comments to the Author

Reviewer #2: (No Response)

Reviewer #3: Manuscript definitely improved. Congratulations to the authors for the very good work. deserving pubblication

7. PLOS authors have the option to publish the peer review history of their article (what does this mean?). If published, this will include your full peer review and any attached files.

Reviewer #2: No

Reviewer #3: No

---

## [Editor Report · Acceptance letter]

15 Feb 2024

PONE-D-23-20202R1 

PLOS ONE

Dear Dr. Tona, 

I'm pleased to inform you that your manuscript has been deemed suitable for publication in PLOS ONE. Congratulations! Your manuscript is now being handed over to our production team.

Kind regards, 

on behalf of

Prof. Vincenzo Lionetti 

Academic Editor

PLOS ONE